# Pareidolia in a Built Environment as a Complex Phenomenological Ambiguous Stimuli

**DOI:** 10.3390/ijerph19095163

**Published:** 2022-04-24

**Authors:** Chen Wang, Liangcheng Yu, Yiyi Mo, Lincoln C. Wood, Carry Goon

**Affiliations:** 1Intelligence and Automation in Construction Fujian Province Higher-Educational Engineering Research Centre, College of Civil Engineering, Huaqiao University, Xiamen 361021, China; wch@hqu.edu.cn (C.W.); 20014086044@stu.hqu.edu.cn (L.Y.); 2College of Civil Engineering, Huaqiao University, Xiamen 361021, China; gooncarry@hotmail.com; 3Department of Management, University of Otago, Dunedin 9054, New Zealand; lincoln.wood@otago.ac.nz

**Keywords:** pareidolia, illusion, misperception, ambiguous stimuli, built environment

## Abstract

Pareidolia is a kind of misperception caused by meaningless, ambiguous stimuli perceived with meaning. Pareidolia in a built environment may trigger the emotions of residents, and the most frequently observed pareidolian images are human faces. Through a pilot experiment and an in-depth questionnaire survey, this research aims to compare built environmental pareidolian phenomena at different time points (6 a.m., 12 p.m., 2 a.m.) and to determine people’s sensitivity and reactions towards pareidolia in the built environment. Our findings indicate that the differences in stress level do not influence the sensitivity and reactions towards pareidolia in the built environment; however, age does, and the age of 40 seems to be a watershed. Females are more likely to identify pareidolian faces than males. Smokers, topers, and long-term medicine users are more sensitive to pareidolian images in the built environment. An unexpected finding is that most pareidolian images in built environments are much more easily detected in the early morning and at midnight but remain much less able to be perceived at midday. The results help architects better understand people’s reactions to pareidolia in the built environment, thus allowing them to decide whether to incorporate it appropriately or avoid it consciously in building design.

## 1. Introduction

Pareidolia is an illusion caused by ambiguous stimuli [1], and the ambiguous forms are perceived as visual objects with meaning. Pareidolia is very common and phenomenological, for example, the visual illusions in dementia with Lewy bodies (DLB) [2]. Pareidolia is a phenomenon where an observer can feel significance from a vague and random stimulus [3]. Many different subjects may appear as pareidolia, but based on previous studies, the most frequent subjects that appear as pareidolia are human faces [4,5]. A newborn baby can recognize faces and human expressions, which means that the human brain may be sensitive enough to detect face-like patterns at birth. Some well-known examples of pareidolia include: seeing the face of Jesus Christ on a potato chip, a cinnamon bun with the face of Mother Teresa, the surface of a grilled sandwich showing the face of the Virgin Mary [6], Satan appearing in the smoke of 9/11, and a devil seen in the Queen’s hair of a 1954 Canadian banknote [7]. The existence of pareidolia could be because of many reasons. In psychology, pareidolia is a partial illusion, and it happens in the condition of low luminance [8,9]. While in neuropathology, the existence of pareidolia is unintentional, and it is a random phenomenon [10]. Normally the pareidolian images received by the human brain are incomplete, but then the brain automatically uses built-in knowledge and the data gathered from previous experiences to fill in the missing parts, generating a complete interpretation that produces a coherent picture [11,12]. From the religious perspective, paranormal believers are more likely to perceive the ambiguous stimuli as face-like patterns due to the sacralization of mythological characters [13].

Pareidolia frequently occurs in the architectural design of house envelopes and facades [14]. In the history of architecture from different cultures, faces frequently occur as decorative. The rock face is one natural formation that often seems to contain a human face [15]. Observers have emotional responses towards those house envelopes they perceive as having face-like patterns [14]. Human faces are the most frequent subject of visual illusion and pareidolia, according to previous studies such as [4]. Some house envelopes consist of a leading and outstanding pattern that can be justified as a human face. Pareidolia phenomena in the built environment may trigger emotional reactions such as happiness, scariness, anxiety, and depression. Therefore, there is a need for building designers to identify those key elements causing pareidolia to prevent negative impact.

Pareidolia is a term that originated from Greek [14]. Basically, the term “pareidolia” is the combination of “para” (para = beside or beyond) and “eidos” (images, appearance, looks), which describes the tendency of the human visual system to extract patterns from noise [8]. In 1885, the Russian psychiatrist Victor Kandinsky (1849–1889) introduced the terms “Pareidolie” and “Nebenbildwahrnehmung” to express a partial visual illusion in which given objects are perceived as different objects, or human faces are precisely and consistently perceived as someone else’s, such as intermetamorphosis syndrome [8].

Pareidolic illusion is another term for pareidolia because it differs from ordinary illusion. Ordinary illusion is a lack of perceptual clarity provoked by intense emotions. When a common illusion becomes more complex and detailed, it will increase the intensity of the pareidolic illusion [10]. Pareidolia is also a form of apophenia, which was first described in terms of psychosis but now is regarded as a tendency in healthy people and could explain or inspire associated visual effects in arts and graphics [14]. According to Dyer [16], the observer’s ability to perceive any random and vague stimuli such as a face is considered pareidolia. Moreover, pareidolia is the illusory perception of a well-known structure such as an animal or a human face, even though no human face or animal exists [17]. There are many examples of pareidolia in various aspects. One example of pareidolia in planetary landforms is the man on the moon or faces and animals in the clouds. An example of pareidolia in a terrestrial object is a face in a tumor ultrasound [17]. Human faces are the most frequently observed subject of visual illusion because of the social importance of faces and our delicate ability to process them [18].

### 1.1. Pareidolia in Psychology and Neuropathology

Pareidolia is a psychological phenomenon that perceives a dedifferentiated sensory stimulus as indicating a familiar object or structure such as a face [17]. Pareidolia causes misperceptions of unreal and unrelated patterns when receiving the vague stimulus, while the stimulus can exist as a glimpse at an unstructured background [18]. There is a consensus that people who display pareidolia have mental insight into the phenomenon, but he or she knows that it is not real [10]. Pareidolic illusion is differentiated from the regular illusion due to the inner impulse. The inner abnormality is not enough for its occurrence of illusion, and an exterior impulse needs to be added, which is why pareidolia is only a partial illusion [8]. Normally, illusion frequently occurs in high luminance, but pareidolia occurs in an opposite condition [9]. In sensory deceptions, the neuropsychological substrate of pareidolia is apophenia. Nowadays, the term apophenia is used in a looser sense, related to perception or psychiatric disease and an excess of perceptual or heuristic sensitivity leading to the judgment of patterns or random connections [8].

In psychopathology, pareidolia is the image seen from shape. It is argued whether the pareidolia phenomenon is non-diseased, voluntary, playful, diseased, unintentional, or distressing [10]. Pareidolia is an automatic phenomenon that occurs rapidly, insensibly, compulsorily, and free of capacity; therefore, humans cannot realize that the perception is misled by pareidolia [11]. However, Liu et al. [18] argued that pareidolia is not merely imaginary because it has a physical reality basis. Since a pareidolian image actually does not contain faces, it needs the interpretive power of the human brain as a substantial tool to detect and connect the faded face-like features to match with the internal face representation [18]. Human experiences are normally filtered through notoriously unreliable senses, so it is impossible for us to really know the truth about the world around us [19].

For the objects containing patterns of two dots and a line segment, the chance for humans to see faces is high because the two dots and line segments appear to the pattern of a human face with eyes and a mouth, which is the most common pareidolia [14]. However, neuropathological and psychiatric conditions can have a negative impact on the ability to identify facial expressions of sensation [14]. Human eyes can distinguish brightness differences in a range of 1–100, which means human eyes’ dynamic contract is high. The human eye is a brilliant detecting system that will undergo both physical and chemical processes from the view and send it to the brain. Though the outcome of the brain–eye system is perception, the brain does not always expound the information of the object seen in the real perception [11]. The most suspect human perceptual apparatus is sight because human eyes do not see a centralized field, while the saccadic movement of the eyes stores the image from detail to detail and is perhaps used later for data recall by forming a whole image [19]. The rhizome structure in the brain has many chemical reaction saccades for the first time of seeing, and it will leave many impressions unprocessed, disordered, and unknown. Since the human brain is a complex structure, the percipient information received in real-time will be eliminated, expanded, rearranged, and codified to form a common and logical layout of the external world. According to Wertheimer and Riezler [20], the Gestalt theory states that visual perception is the effect of the relationship among the objects observed instead of the simple add-up of the elements seen. As a result, the misperception of the shape and color of an object may be caused by the complicated and fast processing work of the brain and lead to illusion and pareidolia.

The brain is the most complex part of the human body, and one of the brain components in charge of face perception is the fusiform gyri [21]. Fusiform gyri shows higher activation when a facial expression is shown on the subject [22]. The face-sensitive neurons in nonhuman primates’ inferior temporal cortex show the same selectivity for face-like object configurations [23]. Even a newborn baby can undergo face recognition and perform across view change [24]. According to Hoback [3], a 3-month-old baby already starts learning to identify the mother’s face, while when the baby grows to four to nine months, the baby can distinguish several facial expressions such as happiness, fear, anger, surprise, or sadness. Moreover, detecting faces from the environment was the survival intuition of humans to ensure vigilance towards danger in the surrounding area [25].

### 1.2. Pareidolia in Religion

A German neurologist and psychiatrist, Klaus Conrad (1905–1961), promoted that pareidolia has a bearing on an even wider range of illusory phenomena, including the discernment of religious themes such as the faces of the Virgin Mary and Jesus [8]. Paranormal believers frequently have perceptual illusions in ambiguous visual stimuli [1]. Even though both paranormal believers and non-believers have the same ability to detect face perception, non-paranormal believers have less liberal response bias than paranormal believers [26], which is probably because paranormal believers perceive ambiguous stimuli as face-like patterns more easily. There is an interrelationship between illusory agency detection and paranormal belief in the studies regarding schizotypy and schizophrenia [13].

Both schizotypy and schizophrenia can cause pareidolia. The schizotypal personality sometimes comes with delusional beliefs, abnormal perceptual experiences, and magical thoughts [27]. According to Galdos et al. [28], a person with a schizotypal personality can sense meaningful patterns in random meaningless noise, while a person with schizophrenia is more able to link unrelated events. By using the moving triangle task, the person with schizotypal can gain meaning from geometrical images that move randomly [13]. Furthermore, some natural landscape elements are sacralized in the three mythological characters: Pan-Gu, Fu-Xi, and Shen-Nong. All these three mythological characters belong to the Chinese culture’s formative period. The three humanized mountains were sacralized to identify these three outstanding Pareidolian Images in their origin. In the case of the commercial aircraft that destroyed New York City’s World Trade Center on 9/11/01, many protestants saw demons and devil faces in the smoke, and they believed that the crisis destroyed the current seat of power of Satan living in the American financial institutions [29]. Pareidolia extends the illusionary connections between landscape and the mythological characters.

### 1.3. Present Study

Pareidolia exists in the built environment, such as landscapes, buildings, and furniture. Many pareidolian faces occur as ornaments or adornments in the architectural history of different cultures [14]. Some house envelopes consist of a leading pattern that can be justified as a human face. The facial expressions perceived in houses can trigger an emotional reaction; thus, the techniques of using pareidolia to produce positive emotions are applied in house design [3]. Experimental studies on the perception of illusory agency discovered that the tendency to sense human faces among random noise is high, which is more than 40%, especially in low information content [17]. There are interrelationships between pareidolian faces and aesthetic value [30]. Though there are many kinds of stimuli able to trigger the emotion of observers, a human face is the most effective one [5]. Examples of pareidolia in the landscape are faces in trees, faces on mountains, faces on stones, the man on the moon, and faces in clouds [18]. Pareidolia sometimes appears on house facades, windows of buildings, doors of buildings, electrical outlets, cupboards, and chairs [14]. Previous research validated the relationship between demographics and delusional disorder using questionnaire surveys and pilot experiments [3,14]. Mental anomalies may be the source causes of many tragedies and incidents [31]. In addition to job pressures, lifestyles and other aspects may affect the mental state to varying degrees. In fact, everyone will be in different mental states at different times of normal life. In general, people are in blurred thinking status when they wake up, while high brain activity during the day will leave people exhausted at night, and the brain activity time is negatively correlated with the human spirit [32]. Negative spirit status increases the possibility of pareidolia [33]. Therefore, this research compares built environmental pareidolian phenomena at different spirit state time points (6 a.m., 12 p.m., and 2 a.m.) with the purpose of determining people’s sensitivity and reactions towards pareidolia in a built environment.

## 2. Materials and Methods

We applied a mix of qualitative and quantitative approaches in this study. We combined a pilot experiment and a questionnaire survey to triangulate the data collected in this study. The pilot experiment is an explanatory and descriptive method to categorize the phenomena of pareidolia in the built environment, which was then used to structure the questionnaire survey and provide readers with a better understanding of pareidolia. The qualitative approach relies on evidence rather than frequency to illuminate issues and uncover possible explanations.

### 2.1. Participants

#### 2.1.1. Pilot Experiment

According to Liu [18], 20 participants (10 females and 10 males, aged from 18 to 60) were randomly recruited from society for the pilot experiment and were offered financial remuneration for participation. The participants mainly consisted of five age groups: 11–20, 21–30, 31–40, 41–50, and 51–60. Four participants per age group. All participants had normal or corrected vision.

#### 2.1.2. Questionnaire Survey

The anonymous questionnaire survey was distributed to 500 people ranging from 11 to 60 years old through the post. In this study, the power was set to 0.8, and the threshold for significance (α) was set to 0.05. According to the power analysis, if we wanted an 80% probability of correctly rejecting the null hypothesis, we needed a sample size of at least 217. A total of 228 valid forms were returned and assessed, representing a 45.6% response rate. The demographic details of the respondents are shown in Table 1. The percentage of females at 59.6% was slightly higher than that of males at 40.4%. The questionnaire survey targeted five age groups: 11–20 years old (6.6%), 21–30 years old (75.4%), 31–40 years old (6.6%), 51–60 years old (6.1%), and 41–50 years old (5.3%). Approximately 72.8% of respondents had a bachelor’s degree, followed by 10.1% with secondary education, 9.2% with a diploma/advanced diploma, 5.3% with primary education, 2.2% with post-graduate degree/professional level, and 0.4% without formal education. In total, 49.6% of respondents considered their current job stressful, and 50.4% did not. There were 62.7% single respondents and 37.3% married. Moreover, 55.7% of respondents normally went to bed after 12 a.m., and 34.6% of respondents did not have any habits such as smoking, alcohol abuse, staying up late, or chronic diseases.

### 2.2. Stimuli

In total, 26 pareidolian images were used for this pilot experiment. The images were obtained from building photos taken on-site and were presented on paper with a picture size of 7.5 × 7.5 cm (H × W). A pareidolian face consists of a facial outline, left and right eyes, and a mouth [30]; thus, the images used in the experiment had similar characteristics.

### 2.3. Experimental Setup and Procedure

#### 2.3.1. Pilot Experiment

In the pilot experiment, 20 participants were required to observe the 26 images provided by the researcher during the morning, midday, and midnight and share their feedback. Each image was observed as an individual experiment by 20 participants. Their feedback was then analyzed using content analysis and structured through tabulation.

#### 2.3.2. Questionnaire Survey

Sensitivity in this study refers to a change in pareidolia’s likelihood when one factor changes. The questionnaire survey determined sensitivity and reactions towards pareidolia in a built environment. There were 69 questions in the questionnaire form. Section A focused on the respondents’ background, such as age, gender, occupation status, education level, lifestyle, and stress level (“Gender”: male, female; “Age group”: 11–20, 21–30, 31–40, 41–50, 51–60; “Educational Level”: no formal education, primary education, secondary education, certificate/diploma/advanced diploma level, degree, post-graduate degree/professional level; “Current job stress level”: stressful, not stressful; “Current status”: single, married; “Lifestyle”: smoking, alcohol abuse, sleep after 12 a.m., long-term medicine user). Section B focused on their sensitivity and reactions towards pareidolia in different spaces in the building. The questions offered a 5-point Likert-type scale for respondents to rank. First, they decided whether the displayed stimuli was a face (“Can you identify a face in Figure?”: yes, no), then they evaluated their sensitivity to face identification (“Can you easily identify a human face in the Figure?”: 5-very highly likely, 4-highly likely, 3-likely, 2-unlikely, 1-denote) and emotions when a face had identified (“What is your reaction when you see Figure?”: scared, depressed, funny, happy, no reaction). Investigating people’s emotional feedback determines what type of architectural design is more likely to be acceptable rather than radical when people generate pareidolia. The specific content of the questionnaire is presented in the Appendix A.

Two software were applied in the analysis: Statistical Package for the Social Sciences (SPSS) Version 22 and Smart PLS 2.0. Structural Equation Modeling (SEM) was used to test the theoretically supported model.

## 3. Results and Discussion

### 3.1. Findings in Pilot Experiments

Data collected from the pilot experiments were used to compare the participants’ sensitivity and reactions toward pareidolian images at different time points (6 a.m., 12 p.m., and 2 a.m.). There were 26 experiments conducted among the 20 participants, and the results are tabulated in Table 2. In general, the participants had very similar sensitivity and reactions towards these pareidolian images. Most participants could identify human faces at 6:00 a.m. in the morning and 2:00 a.m. at night, but not at 12:00 p.m. midday. Most participants felt scared when looking at the pareidolian images at 2 a.m.

According to the feedback from pilot experiments, face identification in the built environments was easier at midnight and in the morning. Especially at midnight, many participants described the faces as scared, whereas they only considered those images as normal pareidolian faces at other time points (6:00 a.m. and 12:00 p.m.). Therefore, the results suggested that pareidolia was more likely to occur when people were exhausted. In other words, when people are fatigued, it will be easier to identify pareidolian faces. Face identification was not significant at 12 p.m., probably because of a higher level of relaxation in brain activity; thus, it did not reach exhaustion at that time. Results from partial images’ feedback showed people reacted differently between morning and midnight when identifying a face in Pareidolian Image 2, 10, 11, and 13. The people’s reactions to images oscillated from “Funny” in the morning to “Scary” at midnight. The majority of pareidolian images reactions were “Scary”, “Neutral”, and “Creepy”. As a result, the pilot experiments revealed that people were more likely to identify pareidolian faces at midnight and in the morning than at midday, and people reacted more negatively to pareidolian images at night.

### 3.2. Results of Questionnaire Survey

The Cronbach’s Alpha of the questionnaire survey was 0.899, showing a high internal consistency and high reliability. In the study, factor analysis was first used to screen the images and eliminate the image results that could not be well classified by factors. The factor analysis' eigenvalue was set at 0.9, and the factor loadings of the original 21 images with an absolute value greater than 0.3 were considered acceptable. There were two criteria for screening images: first, one image had a large loading on two or more factors (factor loading > 0.4), and the difference between the absolute value of the factor loading was less than 0.3; and second, only one image under a factor had factor loading. Only one image could be removed per screening and required reanalysis. The purpose of factor analysis is to integrate highly correlated factors to form new factors. The final KMO value was 0.938 (KMO value > 0.6), and the significance of the Bartlett sphericity test was 0.000 (*p* < 0.05), indicating that the final questionnaire had excellent validity, and the last retained image was suitable for factor analysis. We used factor analysis to remove pareidolian images 1, 4, 5, 6, 7, 8, 9, 11, 13 and 21; the purpose of each image was to examine the effect of demographic information on pareidolia; thus, we could integrate the deleted images into one factor through factor analysis. The final new factor extracted the results from the majority of the images; thus, it could well represent the overall results of the returned questionnaire. We named this factor the pareidolia questionnaire result and used it for multivariate analysis. The image factor loadings in the pareidolia questionnaire result are shown in Table 3.

Table 4 show the results of effect analysis between the demographic information in the pareidolia questionnaire results. Among them, age, gender, and lifestyle as a single factor showed a significant impact on pareidolia. Many two–factor between–subject effects also significantly influenced pareidolia identification, including Age*Gender (*p* = 0.036 < 0.05), Age*Lifestyle (*p* = 0.000 < 0.05), Gender*Lifestyle (*p* = 0.000 < 0.05) and Job Stress*Lifestyle (*p* = 0.000 < 0.05). It was worth noting that the three–factor and four–factor interaction effects did not show a significant influence on pareidolia identification.

Figure 1 show the effect of significant two–factor between-subject effects on pareidolia identification. Two–factor estimated marginal means can not only show between–subject effects but also reflect the influence of a single factor on pareidolia. Figure 1a show the differences in pareidolia identification between males and females in different age groups. From the overall trend, females were more likely than males to identify pareidolian faces. Especially in the 21–40 age group, females performed much better than males in pareidolian face identification. However, gender had no significant influence on face identification in the 41–60 age group. Compared with gender, age had a significant impact on pareidolia. The performance of pareidolian face identification improved with increasing age. It seemed that 40 years old was a watershed since the 41–50 age group displayed a significant difference from the 31–40 age group. When the age exceeded 40 years old, the performance of pareidolian face identification showed a steady trend.

Figure 1b show the differences in pareidolia identification between males and females with different lifestyles. Lifestyle significantly influenced pareidolia identification. The two–factor between-subject effects of lifestyle with age, gender, and job stress all demonstrated significant influence. Figure 1b,c both showed that long-term medicine users were more likely to identify pareidolia. Topers and smokers behaved similarly, but both influenced pareidolia identification to a large extent. Sleeping late had little influence on pareidolia recognition. However, people without these lifestyles seemed to be less prone to pareidolia. Figure 1b show the differences in pareidolia between males and females with different lifestyles. Male smokers and long-term medicine users were more likely to identify human faces than females. However, females who abused alcohol, slept late and did not have these lifestyles were more likely to identify human faces than males. This also confirmed that females were more likely to identify human faces than males in daily life. Figure 1c show the between–subject effects between job stress and lifestyle. There was little difference in pareidolia identification between long-term medicine users, smokers, and people without these lifestyles, whether they had job stress. Among late sleepers, people with job stress were more likely to identify human faces. Conversely, topers without job stress appeared to easily identify human faces.

Age and lifestyle were two factors that significantly affected pareidolia; the between-subject effect remained significant for pareidolia identification. From Figure 1d, people in the 41–60 age group had significantly higher pareidolia identification between different lifestyles than those in the 11–30 age group. Other lifestyle differences were revealed to not be significant in the 41–60 age group, except for long-term medicine users. However, in the 11–40 age group, long-term medicine users had a significant impact on pareidolia. Topers and smokers had a large degree of sensitivity to identifying human faces in images. Other lifestyles did not show a significant influence on pareidolia.

### 3.3. PLS–SEM Analysis

This study used the PLS–SEM model to investigate the causal relationships between pareidolian images, pareidolia sensitivity, demographics, and pareidolian reactions. Since PLS modeling focuses on prediction and makes no assumptions regarding data distribution, residuals, or parameters of the observable variables, only generic linear regression requirements must be met. PLS modeling is more reliable than LISREL in analyzing the causal relationship between pareidolia factors. Figure 2 show the initial PLS–SEM model of this study.

To verify the rationality of the observed variables among the four latent variables, we conducted a reliability analysis on the latent variables. The reliability test was regarded passed if the Cronbach’s Alpha value and CR value in the internal consistency reliability test were both greater than 0.70. Table 5 show the initial latent variable reliability test results. The CR values of “Demographic” and “Can identify phenomenon of pareidolia” were both greater than 0.7, but the Cronbach’s Alpha value failed the test, indicating that the observed variables need to be modified or removed. Observed variables with variable loadings less than 0.5 were considered to lack correlation; thus, we removed “Working” and “Education” from “Demographic” and “Pareidolia 2” and “Pareidolia 21” from “Can identify the phenomenon of pareidolia”, respectively. The reliability test and correlation test of the revised PLS–SEM model both met the standard, indicating that the selected data had good reliability. Table 6 show the convergent validity test results of the PLS–SEM model. The AVE values of revised models were all greater than 0.5, suggesting that the revised model had better construct validity.

According to the PLS–SEM model path analysis in Figure 3, “Human face identification” (0.675), “Can identify phenomenon of pareidolia” (0.271), and “Demographics” (0.374) all had a direct impact on “Reaction”, and “Human face identification” (0.675) witnessed the most significant influence on “Reaction”; this may infer that people found it easier to identify pareidolian faces in the built environment, resulting in stronger emotional reactions. “Demographics” showed no significant influence on both “Human face identification” (0.163) and “Can identify the phenomenon of pareidolia” (0.256). As demographic factors such as age and lifestyle changed, “Human face identification” and “Can identify the phenomenon of pareidolia” also experienced some variations. “Can identify the phenomenon of pareidolia” had a moderate impact on “Human face identification”; this may indicate that people were more likely to identify human faces in a built environment under an illusion. As a result, “Human face identification” was the most significant factor in influencing people’s reactions to the built environment. In addition, “Can identify the phenomenon of pareidolia” and “Demographics” both had a greater influence on pareidolian reaction through the indirect influence of “Human face identification”.

### 3.4. Discussions

Consistent with the results of many studies, gender influenced the sensitivity and reactions towards pareidolia in a built environment, which might be explained by Pavlova et al. [34], who claimed that females are more likely to detect faces in arrangements of food on a plate. Age had a significant influence on pareidolia sensitivity. According to the single factor analysis, respondents’ sensitivity to identify faces increased with age in the 11–40 age group, and they were more likely to have pareidolia than those over 40. This might indicate that the age of 40 is a watershed for pareidolia. However, there were significant differences between males and females in the 21–40 age group but non-significant differences in other age groups. Job stress seemed to have no significant impact on sensitivity to pareidolia in the built environment, which was not fully in line with [11]. Although the between-subject effects between job stress and lifestyle demonstrated a significant influence on pareidolia identification, the experimental results revealed that this influence mainly relied on the lifestyle factor. This is probably because pareidolia is an automatic phenomenon that occurs rapidly, insensibly, and compulsorily, and people do not realize their perception is misled by pareidolia [30]. Different lifestyles had some impact on the sensitivity and reactions towards pareidolia. This could be explained by Chalup et al. [14], who claimed that neuropathological and psychiatric conditions could affect the sensitivity and reactions to identify facial expressions of sensation. In the current research, smokers, topers, and long-term medicine users tended to be more sensitive to pareidolian images in the built environment. Combining gender factors, long-term medicine users and smokers in males were more sensitive than females in face identification. Additionally, females who abused alcohol, slept late, and did not have these lifestyles were more likely to identify faces than males. This finding was consistent with previous findings that females are more likely to identify faces in pareidolian images [34] since sleeping late and having none of these lifestyles is common of most people nowadays. In addition, the between-subject effects of age and lifestyle both showed a significant influence on pareidolia. The 41–60 age group were more likely to identify faces than those in the 11–40 year age group with different lifestyles. However, long-term medicine users in different age groups were more likely to identify faces in pareidolian images. Among the respondents who slept late and did not have these lifestyles, the 41–60 age group differed significantly from those in the 31–40 age group in pareidolian face identification.

Through the path analysis of the PLS-SEM model, “Human face identification” significantly influenced people’s reactions to pareidolia in the built environment, and demographic information also showed a moderate influence on pareidolian reactions. Pareidolian human faces appearing in buildings may cause positive or negative effects on human life. Some may lead to happiness and relaxation, but more lead to feelings of creepiness and depression. The facial expressions perceived in house designs through pareidolia can trigger emotional reactions based on the time of day [3].

## 4. Conclusions

We can categorize Pareidolia in a built environment into unintentional and intentional. The sensitivity and reactions towards pareidolia in a built environment are spontaneous and influenced by the age, gender, and lifestyles of observers. The majority of the respondents chose to change the design layout when they detected pareidolia in the built environment, including those living in the built environment with intentional pareidolian design elements. Residents rarely like any pareidolian faces in buildings, whether intentionally or unintentionally. The difference in gender influences the sensitivity and reactions towards pareidolia in a built environment because females have a stronger capability in face identification. The age of the respondents has a significant influence on pareidolia, and the age of 40 seems to be a watershed. Job stress has little impact on sensitivity to pareidolia in a built environment, but the between-subject effects of job stress and lifestyle have a considerable impact on pareidolia due to the dominance of lifestyle. The sensitivity to pareidolia varies depending on one’s lifestyle. Smokers, topers, and long-term medicine users are more sensitive to pareidolian images in the built environment. Similar results are observed even when gender and age factors are taken into account. The identification of the pareidolian face in the built environment significantly influences pareidolian reactions and sensitivity. In addition, the pareidolian human faces appearing in buildings may cause positive or negative effects on human life. Some may lead to happiness and relaxation, but more lead to creepiness and depression. The facial expressions perceived in house design through pareidolia can evoke the emotion of the observers. As a limitation, a finding that remained unexpected is that most of the pareidolian images in the built environment are much more easily detected during early morning and at midnight but remain much less able to be perceived at midday. As a result, further and more in-depth studies are required, especially multi-disciplinary studies, to address this issue satisfactorily.

## Figures and Tables

**Figure 1 ijerph-19-05163-f001:**
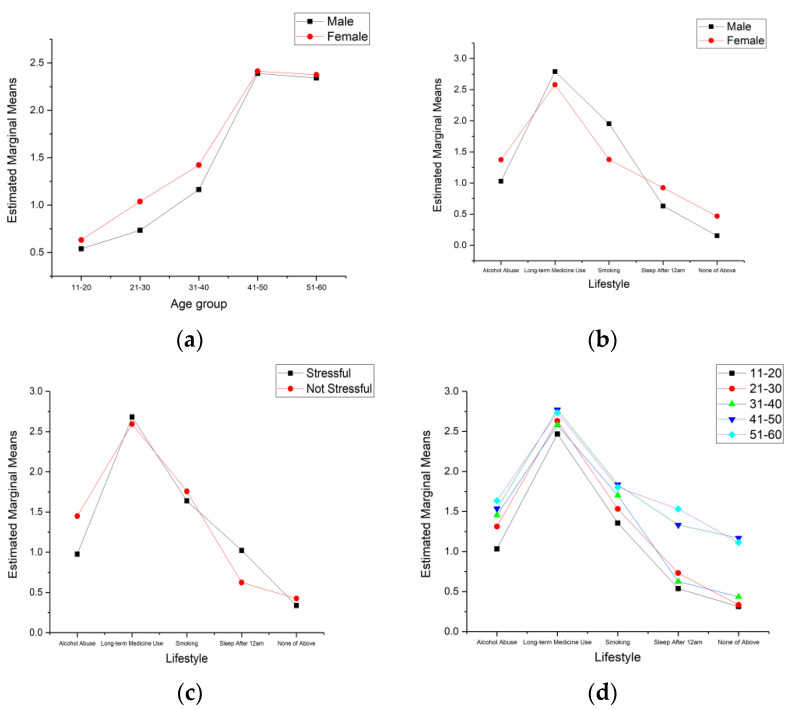
Two–factor Estimated Marginal Means of pareidolia questionnaire result. (**a**) Gender and age group; (**b**) Gender and lifestyle; (**c**) Job stress and lifestyle; (**d**) Age group and lifestyle.

**Figure 2 ijerph-19-05163-f002:**
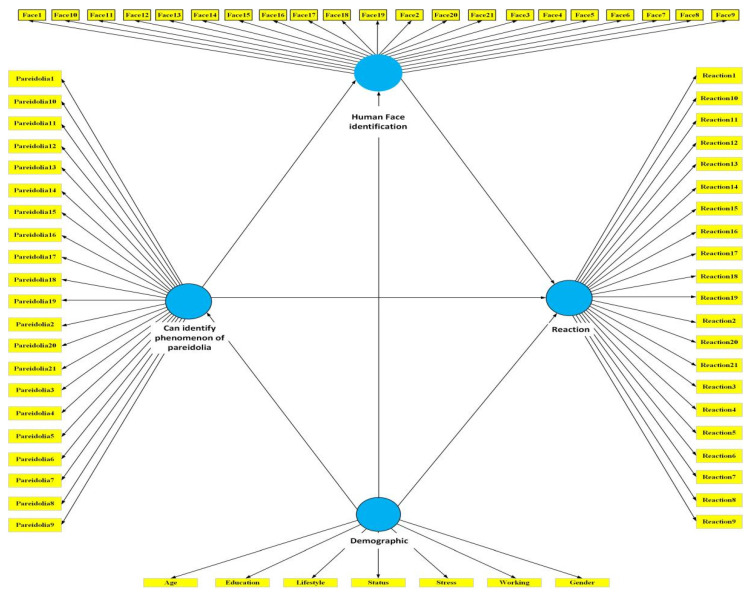
Initial PLS–SEM model of demographic data and sensitivity to pareidolia in the built environment.

**Figure 3 ijerph-19-05163-f003:**
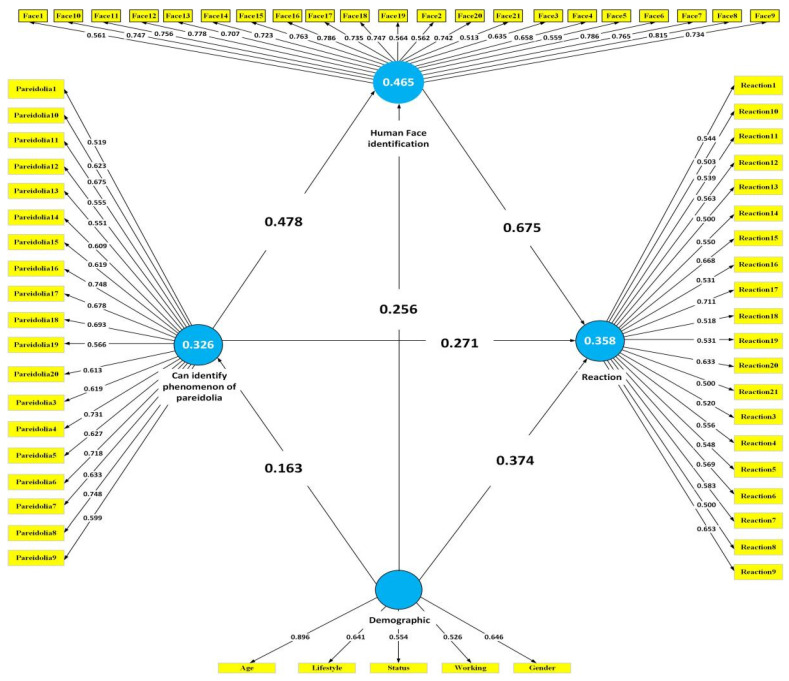
SEM tests between demographic data and sensitivity to pareidolia in the built environment.

**Table 1 ijerph-19-05163-t001:** Demographic details of respondents.

Demographic Categories	Category Breakdown	Frequency	Percent (%)	Cumulative Percent (%)
Gender	Male	92	40.4	40.4
	Female	136	59.6	100
Age Group	11–20	15	6.6	6.6
	21–30	172	75.4	82
	31–40	15	6.6	88.6
	41–50	12	5.3	93.9
	51–60	14	6.1	100
Education Level	No formal education	1	0.4	0.4
	Primary education	12	5.3	5.7
	Secondary education	23	10.1	15.8
	Certificate/diploma/advanced diploma level	21	9.2	25
	Degree level	166	72.8	97.8
	Postgraduate degree/professional level	5	2.2	100
Current Job Stress Level	Not stressful	113	49.6	49.6
	Stressful	115	50.4	100
Current Status	Single	143	62.7	62.7
	Married	85	37.3	100
Lifestyle	Smoking	8	3.5	3.5
	Alcohol abuse	7	3.1	6.6
	Sleep after 12 a.m.	127	55.7	62.3
	Long-term medicine user	7	3.1	65.4
	None of above	79	34.6	100

**Table 2 ijerph-19-05163-t002:** Tabulation of experiments results.

Pareidolian Image	Phenomenon	Reaction
6 a.m.	12 p.m.	2 a.m.	6 a.m.	12 p.m.	2 a.m.
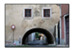	Face is detected	Normal building	Face is detected	Depress and sad	Neutral	Creepy and scary
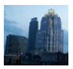	Looked like a blue monster	Normal building	Eyes are detected	Scary	Neutral	Creepy
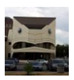	Cartoon-like face is detected	Fish mouth and eyes are detected	Fierce face is detected	Funny	Neutral	Creepy
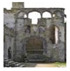	Face is detected	Normal building	Face is detected	Scary	Neutral	Scary
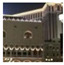	The face is not really obvious	Face is detected	Normal building	Neutral	Happy and cute	Neutral
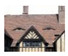	Face is detected	Normal building	Fierce face is detected	Scary	Scary	Creepy
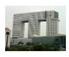	The face is not really obvious	Normal building	Normal building	Neutral	Neutral	Neutral
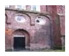	Face is detected	Normal building	Fierce face is detected	Neutral	Neutral	Creepy
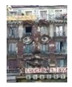	Face is detected	Face is detected	Face is detected	Neutral	Neutral	Neutral
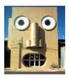	Face is detected	Face is detected	Face is detected	Funny	Funny	Scary
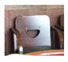	Face is detected	The face is not really obvious	Face is detected	Funny	Funny	Scary
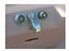	Face is detected	Face is detected	Face is detected	Scary	Funny	Scary
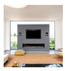	Face is detected	Face is detected	Fierce face is detected	Funny	Funny	Scary
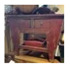	Face is detected	The face is not really obvious	Fierce face is detected	Neutral	Neutral	Scary
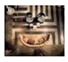	Face is detected	Face is detected	Fierce face is detected	Scary	Scary	Scary
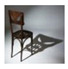	The face is not really obvious	The face is not really obvious	The face is not really obvious	Neutral	Neutral	Neutral
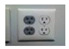	The face is not really obvious	The face is not really obvious	The face is not really obvious	Neutral	Neutral	Neutral
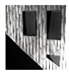	The face is not really obvious	The face is not really obvious	Face is detected	Neutral	Neutral	Scary
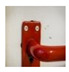	The face is not really obvious	The face is not really obvious	Face is detected	Funny	Funny	Funny
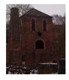	Face is detected	Face is detected	Face is detected	Scary	Funny	Scary
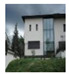	Face is detected	Face is detected	Face is detected	Neutral	Neutral	Neutral
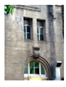	Face is detected	Face is detected	Fierce face is detected	Scary	Scary	Scary
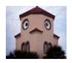	Face is detected	Face is detected	Face is detected	Funny	Funny	Funny
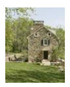	Face is detected	Face is detected	Face is detected	Scary	Funny	Scary
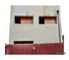	Face is detected	The face is not really obvious	Face is detected	Neutral	Neutral	Neutral
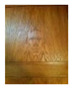	The face is not really obvious	The face is not really obvious	Fierce face is detected	Scary	Neutral	Scary

**Table 3 ijerph-19-05163-t003:** The image factor loadings in the pareidolia questionnaire result.

**Image Number**	2	3	10	12	14	15	16	17	18	19	20
**Factor** **Loading**	0.838	0.782	0.906	0.736	0.771	0.788	0.891	0.581	0.595	0.932	0.894

**Table 4 ijerph-19-05163-t004:** Tests of Between–Subjects Effects.

Source	Type III Sum of Squares	df	Mean Square	F	Sig.
Age	65.413	4	16.353	233.348	0.000
Gender	0.307	1	0.307	4.383	0.038
Job Stress	0.211	1	0.211	3.017	0.084
Lifestyle	1.156	4	0.289	4.124	0.003
Age*Gender	0.609	3	0.203	2.898	0.036
Age*Job Stress	0.010	2	0.005	0.068	0.934
Age*Lifestyle	3.134	7	0.448	6.388	0.000
Gender*Job Stress	0.002	1	0.002	0.025	0.874
Gender*Lifestyle	2.577	3	0.859	12.259	0.000
Job Stress*Lifestyle	3.020	2	1.510	21.544	0.000
Age*Gender*Job Stress	0.001	1	0.001	0.018	0.892
Age* Gender*Lifestyle	0.019	1	0.019	0.268	0.605
Age*Job Stress*Lifestyle	0.025	1	0.025	0.360	0.549
Age*Job Stress*Lifestyle	0.023	1	0.023	0.325	0.569
Age*Gender*Job Stress*Lifestyle	0.059	1	0.059	0.837	0.362

**Table 5 ijerph-19-05163-t005:** Initial latent variable reliability test results.

Structural Reliability	Demographic	Can Identify Phenomenon of Pareidolia	Human Face Identification	Reaction
Cronbach’s Alpha	0.639	0.673	0.819	0.771
CR	0.767	0.716	0.849	0.733

**Table 6 ijerph-19-05163-t006:** Convergent validity test results of revised model.

Convergent Validity	Demographic	Can identify Phenomenon of Pareidolia	Human Face Identification	Reaction
AVE	0.718	0.633	0.762	0.629

## Data Availability

All data, models, and code generated or used during the study appear in the submitted article.

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
