# Peer review of "Pareidolia in a Built Environment as a Complex Phenomenological Ambiguous Stimuli"

_ijerph, 2022, doi:10.3390/ijerph19095163_

Round 1

Reviewer 1 Report

The revision reads very well, albeit it is dense material. The authors made reasonable edits to address my queries/concerns reasonably well, except I do not see where the authors explain how many competing SEM models they tested before showing the model in Figure 2. That is, I assume the Fig 2 model was the best-fitting to their data, although readers do not know what other directional relationships were tested, etc. I think this is important to address for the sake of rigor and completeness. Otherwise I support publication at this point.

Reviewer 2 Report

This manuscript is much improved. I recommend publication, but might perhaps insist on a table presenting factor analysis loadings for the readers clarity. I ultimately leave that decision to the editor. 

Author Response

This manuscript is a resubmission of an earlier submission. The following is a list of the peer review reports and author responses from that submission.

Round 1

Reviewer 1 Report

Review: Pareidolia in Built Environment as Complex Phenomenological Ambiguous Stimuli.

The number indicates the line in manuscript comment refers to.

  1. It may not be appropriate to refer to pareidolia as “hallucination”, particularly since you describe the cognitive processes which generally are also used for the reconstruction of memory.
  2. While appreciating any research further examining pareidolia in context of environment, this statement needs further information. Why is there a need to prevent negative impact of pareidolia in environments? Research on pareidolia is sparse in general, as such, its negative impact is not empirically determined.
  3. This is a highly selective claim based on limited research. Yes, there is a relationship between schizotypal sub-clinical features and paranormal experience related to hypothetical pareidolia experiences. However, the overall relationship here (correlation) is approximately .3, accounting for a 10% overlap at best of schizotypal tendency and what you are inferring. Further, Schizotypal personality disorder per the DSM 5 is at best 3% of the population, whereas pareidolia seems generally distributed among the population as an inherent cognitive quirk or phenomena. NOT all sub-clinical or formal clinical diagnoses would necessarily contain the symptoms you describe as “always comes with”. Contextual references and more research honesty is needed in this section.
  4. It is unclear if the pilot participants are separate from the survey participants. Further, there is no details on how either demographics or sensitivity to or reactions to pareidolia spaces was defined. A large amount of detail is needed here so that replication could be conducted.
  5. As the pictures are provided in a previous table, I am unclear why Table 3 needs to be in this paper. This is particularly true in the sense that a: the rating and method of measurement should have been defined before the results section, and b. results could be summarized here IF they were defined previously in the text.
  6. One significant finding in 23 analyses is not any form of acceptable evidence for a finding to report. 21 other images found no effect. By chance alone, one finding would be significant. Collapse these analyses.
  7. Aside from poor interrelation within the SEM analysis, showing as the other tables do that several of the selected pareidolia images do not seem to elicit any effects, SEM should be its own separate paper, and perhaps the first analysis you conducted to provide some form of structure and validity to your pareidolia images. As your tables and SEM analysis show, there is a large degree of inconsistency here between the images and reactions, which in turn, aside from the issues above, make the entire results section suspect. Your primary variable of pareidolia has not been appropriately modeled or operationalized.

GENERAL COMMENT: All of these tables are not viable, and it is also unclear as to why the authors did not sum their findings to conserve probability in testing. Asides from type I error running rampant across all of these analyses, what is the value of individual analyses for each pareidolia image? No rationale is provided as to why these should be analyzed separately, and the lack of significance in many of the images reinforces that. Further, the authors are creating conditions of “sensitivity” which in fact relate to the viability of their images selected. This serves as a natural confound to the findings, as the authors in the current analysis are really showing us that within all images, a specific subset produces significant results, while the other images do not. This needs to be addressed in detail and may require reperforming analyses. Finally, given the large amount of analyses, and somewhat sparse significance within individual images, I am not convinced that the findings within these specific images are not artifacts due to the massive degree of analyses run.

Summation: No further review is necessary of either the introduction or discussion as there are systemic analysis issues in this paper in terms of operational definitions, details of method, details of procedure, and the analysis method and sheer number of analyses have destroyed the clarity of statical significance. However, this paper’s topic is worthwhile, and it is my hope that the authors will seriously review their design and analysis to clearly present the aggregate findings of their work.

Reviewer 2 Report

Please see my attached report.

Round 2

Reviewer 1 Report

I appreciate the author’s efforts to involve clarity from their previous revision. However, the authors did not choose to alter their analysis in any significant way, and the sheer number of analyses conducted (approximately 1038 significance tests by my count) creates an absolute doubt on a large body of the researcher’s findings (60 analyses at the p < .05 level are probabilistically spurious). As a demonstration, although I am not fond of a Bonferroni correction, would set significance for this bombardment of analysis at .0008, notably rendering most of the findings insignificant.

It remains the case (noting the intercorrelation table) that these images should have first been filtered through factor analysis (or SEM applied as a scaling instrument) in order to create an overall metric of pareidolia faces. The results would have likely removed more than half of the images, and subsequently created a series of several images which could have been combined into one measure. As such, the required analyses become manageable, and not a claim that a few images within 21 images showed significant effects.

Unfortunately, despite the author’s attempts to improve clarity, the data from which this study was conducted has been data-mined to the point that it would be deceptive to the public to present data in this manner, due to the rampant Type I error created from this many analyses. Further, the sheer amount of data presented in tables is not practically useable. I strongly recommend the authors remove most of these analyses, conduct their scaling first to create a pareidolia metric, and then conservatively conduct analyses based on a demonstratable body of interrelated pareidolia images.

Reviewer 2 Report

This revision much improves on the original version. However, many ambiguities remain that must be rectified.

First, the paper details two separate studies although there is virtually no delineation between them. This must be remedied, which requires some reformatting per the journal's style guidelines.

Second, there is no explanation, much less justification, for the sample sizes used here. For instance, Study 1: Experiment had only 20 people. Why so few? Study 2: Survey invited 500 and ended up with 228 valid forms. Was any power analysis done to establish sample sizes here?

Third, the Method/Materials section(s) remain unclear. No summary age data is given for the Participants section (the data for Study 2 on this point is given later in another section for some reason).

Fourth, the questionnaire used was not reproduced in an Appendix or even available via a hyperlink. Thus, readers don't have a strong sense about the directions and questions. There is also no data attesting to its psychometric quality given under Method/Materials (again, an alpha coefficient is oddly reported in the Results section). Thus, the Method/Materials section should be reworked per APA style.

Finally, this is not a strict requirement, but one suggestion for the authors to consider that would strengthen their arguments. The survey study might have enough data for a split-sample validation. Namely, the 228 completed forms could be randomly divided in half, with one half serving as an Exploratory Sample for significant effects, whereas the other half is a Validation Sample to immediately replicate any such significant findings. Thus, an immediate replication would look more impressive than what is presented here.

Taken altogether, the revision is closer to a publishable work but not quite there yet.
